# Interactions of COMT and ALDH2 Genetic Polymorphisms on Symptoms of Parkinson’s Disease

**DOI:** 10.3390/brainsci11030361

**Published:** 2021-03-12

**Authors:** Rwei-Ling Yu, Shao-Ching Tu, Ruey-Meei Wu, Pei-An Lu, Chun-Hsiang Tan

**Affiliations:** 1Institute of Behavioral Medicine, College of Medicine, National Cheng Kung University, Tainan 701, Taiwan; lingyu@mail.ncku.edu.tw; 2School of Medicine, College of Medicine, Kaohsiung Medical University, Kaohsiung 807, Taiwan; coolesttaiwaneseever@gmail.com; 3Department of Neurology, National Taiwan University Hospital, College of Medicine, National Taiwan University, Taipei 100, Taiwan; robinwu@ntu.edu.tw; 4Department of Medical Imaging and Radiological Sciences, Kaohsiung Medical University, Kaohsiung 807, Taiwan; u106029051@kmu.edu.tw; 5Department of Neurology, Kaohsiung Medical University Hospital, Kaohsiung Medical University, Kaohsiung 807, Taiwan; 6Graduate Institute of Clinical Medicine, College of Medicine, Kaohsiung Medical University, Kaohsiung 807, Taiwan

**Keywords:** COMT, ALDH2, Parkinson’s disease, bradykinesia

## Abstract

(1) Background: Monoamine neurotransmitters play essential roles in the normal functioning of our nervous system. However, the metabolism of monoamine neurotransmitters is accompanied by the production of neurotoxic metabolites, and inefficient removal of the metabolites has been suggested to cause neurodegeneration. (2) Methods: To examine the effect of reduced activity of catechol-O-methyltransferase (COMT) and aldehyde dehydrogenase 2 (ALDH2) conferred by single nucleotide polymorphisms COMT rs4680(A) and ALDH2 rs671(A) on the symptoms of patients with Parkinson’s disease (PD), a total of 114 PD patients were recruited cross-sectionally and received genotyping for rs4680 and rs671 along with MDS-UPDRS evaluation. (3) Results: We found that patients carrying rs4680(A) had more severe bradykinesia in the upper extremity and rest tremor. Besides, patients carrying rs671(A) had more difficulty maintaining personal hygiene, while patients with genotype rs671(GG) had higher scores in the item “depressed mood.” More importantly, we found the effect of rs4680 to be moderated by rs671 SNP for the symptom of “hand movements.” The detrimental impact of rs4680(A) is more pronounced in the presence of genotype rs671(GG). (4) Conclusions: This study facilitates a deeper understanding of the detrimental effect of reduced activity of COMT and ALDH2 conferred by genetic variation and provides novel insight into the interactions between enzymes metabolizing monoamine neurotransmitters in the pathogenesis of PD.

## 1. Introduction

Parkinson’s disease (PD) is a neurodegenerative disease characterized by progressive denervation in the striatum and substantia nigra, areas critical for dopamine transmission [1,2,3,4]. It can be presented with a wide range of symptoms, broadly categorized as motor and non-motor symptoms. Furthermore, the significant variability in disease presentation, progression rate, and response to treatment among PD patients all indicate the heterogeneous pathogenic mechanisms of PD [5]. Besides, a great variety of mechanisms contributing to the pathogenesis of PD have been proposed, and the numerous pathogenic mechanisms proposed all indicate that the neurodegeneration in PD consists of complex interactions between genetic susceptibility and environmental factors [6]. The fact that the proposed mechanisms fail to explain the entire picture of the pathogenic process of PD also indicates the existence of other mechanisms to be explored.

Dysregulation of the monoamine neurotransmitter has been proposed to be involved in the pathogenesis or clinical manifestations of PD [7,8]. The metabolism of dopamine and norepinephrine begins via either catechol-O-methyl transferase (COMT) or monoamine oxidase (MAO) to intermediate metabolites. When dopamine is initially catalyzed by COMT, 3-methoxytyramine is formed. When initially metabolized by MAO, dopamine is oxidized to 3,4-dihydroxyphenylacetaldehyde (DOPAL) [9,10]. DOPAL can be further metabolized by either ALDH to form 3,4-dihydroxyphenylacetic acid (DOPAC) or alcohol dehydrogenase/aldose reductase to 3,4-dihydroxyphenylethanol (DOPET). Growing evidence has suggested that DOPAL and DOPAC are associated with neuronal death, and timely removal of the neurotoxic metabolites through enzymatic activity is crucial in reducing the neurotoxicity secondary to monoamine neurotransmitter metabolism [10,11]. We hypothesized that reduced COMT activity would cause more monoamine neurotransmitters to be metabolized by MAO, resulting in increased production of neurotoxic metabolites [10,11], DOPAL and DOPAC, and the reduced COMT activity would also hamper efficient metabolism of DOPAC [12,13].

Meanwhile, reduced ALDH2 activity would also cause inefficient metabolism of DOPAL, the accumulation of which also results in neurodegeneration [10]. Furthermore, an imbalance in the pathway metabolizing these neurotoxic metabolites may lead to the accumulation of either DOPAL or DOPAC, and neurotoxicity ensues. In other words, both COMT and ALDH2 inactivity and imbalance between COMT and ALDH2 may cause neurotoxicity.

An Asian-specific ALDH2 single nucleotide polymorphism (SNP), rs671(A) [14], causes the tissue ALDH2 activity of heterozygotes to be reduced to 17% of that of rs671(GG) individuals, whereas the tissue ALDH2 activity of rs671(AA) individuals was too low to be determined [15]. Meanwhile, another nonsynonymous SNP, rs4680, with a moderate difference in the allele frequency across different ethnicities, causes a significant reduction in COMT activity [16]. However, results across various studies investigating the impact of the COMT SNP on individuals suffering from PD diverge, where some studies showed a detrimental impact of the SNP on the motor [17] and non-motor symptoms [18], while some studies showed no association of the SNP on the risk or clinical features of PD [19]. We hypothesized that the interactions with the genetic polymorphism of rs671(A) might be one reason for the diverging results. Moreover, the rs671(A) allele frequency is around 0.226 to 0.255 in Asians [20,21], indicating that 40% of Asians carry the SNP rs671(A), causing reduced ALDH2 activity. According to a Deloitte report in 2017, Asia is expected to be home to more than 60% of the total population older than 65 years old worldwide by the 2030s. Besides, the life span increase of Asians leads to the prediction that PD may affect around 5 million people in the area by 2030, which will account for nearly 60% of the PD population in the world [22]. Therefore, understanding the impact of rs671 and rs4680 polymorphisms and the interaction between the two SNPs on Asian PD patients is crucial.

To validate our hypothesis with potentially significant implications in pathogenesis and manifestation of PD, we explored 1) the impact of inactive COMT and ALDH2 SNPs rs4680(A) and SNP rs671 (A) on the symptoms of PD patients; 2) the impact of interactions between COMT rs4680(A) and ALDH2 rs671 (A) in the motor and non-motor symptoms of PD patients.

## 2. Materials and Methods

### 2.1. Participants and Assessment of Clinical Presentations

This study included 114 patients diagnosed with idiopathic PD according to the MDS clinical diagnostic criteria for Parkinson’s disease [23]. The number of participants recruited for the current study was based on a statistical power analysis performed with Gpower 3.1 according to our previous study’s data investigating the effect of ALDH2 rs671 SNP on the non-motor symptoms of PD patients, in which the effect size was calculated to be 0.801. With an α = 0.05, power = 0.95, and an equal number of participants in each subgroup, the projected sample size needed with the calculated effect size is approximately *N* = 84. These patients were referred from neurologists from the outpatient departments from three medical centers located in three different cities in Taiwan from 2012 to 2019 cross-sectionally. We excluded patients with age of onset before 50 years old. Those with a history of disease duration longer than ten years were also excluded as dopamine replacement therapy may disrupt the natural trajectories of disease progression and clinical presentation. We also excluded patients with the following features: illiteracy, brain surgery history, or any severe systemic disease.

The written informed consent from the participants was obtained before enrollment, following the ethical standards outlined in the 1964 Declaration of Helsinki. All study procedures were approved by the ethical research committee of Kaohsiung Medical University Hospital, National Cheng Kung University Hospital, and National Taiwan University Hospital. All methods were performed according to the approved guidelines. The detailed information of demographic and dopamine replacement therapy and motor severity were obtained from the medical record assessed by the neurologists. Levodopa equivalent dose was calculated according to the protocol previously published [24]. The motor symptoms and severity of the patients were evaluated according to Movement Disorder Society-Unified Parkinson’s Disease Rating Scale (MDS-UPDRS) [25] and Hoehn and Yahr staging criteria [26], respectively.

### 2.2. Genetic Analysis

Peripheral blood leukocytes from the 114 patients were collected for genomic DNA extraction. The genomic DNA of 85 patients was extracted with the Genomic DNA Extraction Kit of Geneaid (Geneaid, New Taipei City, Taiwan), and the genomic DNA of 29 patients was extracted with TANBead Nucleic Acid Extraction Kit. The rs671 and rs4680 SNP genotypes of 73 patients were determined with TaqMan probes and the StepOnePlusTM system with the StepOne software (Applied Biosystems, Grand Island, NY, USA), and the genotypes of the other 41 patients were determined with C2-58 Axiom Genome-Wide TWB 2.0 Array Plate based on the Affymetrix GeneChip platform. The laboratory technicians were blinded to the patient’s demographic characteristics, performed the genotyping, and read the results.

### 2.3. Statistical Analysis

Similar to our previous publications [7,8], proportions were calculated for qualitative variables, and means and standard deviations (SDs) were calculated for quantitative variables. We applied the Kolmogorov-Smirnov test for testing normality. Afterward, we tested quantitative variables with the t-test or Mann–Whitney U test and tested qualitative variables using a chi-square test. We controlled the impact attributable to confounding variables (age, sex, and LED) by applying Quade’s test. Statistical significance was predetermined with an alpha level less than 0.05. The statistical analysis was performed by a commercially available software program (IBM Corp. Released 2012. IBM SPSS Statistics for Windows, Version 21.0. Armonk, NY: IBM Corp.). A regression-based analysis was carried out with PROCESS macro on SPSS [27] to calculate the interaction between COMT rs4680 and ALDH2 rs671 on the symptoms affected by the SNPs. All the associated data not provided within the paper are available on request from Chun-Hsiang Tan.

## 3. Results

### 3.1. COMT SNP rs4680(A) Is Associated with More Severe Motor Symptoms

To examine the effect of COMT SNP rs4680(A) on the clinical presentations of patients with PD, we compared the ratings obtained with MDS-UPDRS between PD patients carrying rs4680(A) allele and patients with the genotype of rs4680 (GG). The allele frequency of rs4680(A) of the recruited patients was 24.6% (56 out of 228). The patients’ genotype frequencies were in accordance with Hardy-Weinberg equilibrium (χ^2^ = 2.115142217, *p* = 0.145849). There were no significant differences in age, gender, age of onset, levodopa equivalent daily dose (LED), entacapone use, and disease severity (Hoehn and Yahr stage, H&Y stage) between the two groups (Table 1).

With the comparable demographic characteristics between the two groups, we found a significant difference in the symptoms of bradykinesia in the upper extremity (Table 2). With Mann-Whitney U test, we found that patients carrying rs4680(A) allele had higher scores in the items of “hand movements” (U = 1191.5, *p* = 0.014) and “pronation-supination movements” (U = 1220.5, *p* = 0.023). To control the potential effect of confounding factors, including age, sex, and LED, we also applied Quade’s test to examine the effect of rs4680(A) on UPDRS. The results are similar: the group carrying rs4680(A) allele had more severe symptoms in the items of “hand movements” (F = 5.888, *p* = 0.017) and “pronation-supination movements” (F = 5.021, *p* = 0.027). Besides, after controlling the effect of confounding factors, the group carrying rs4680(A) allele had more severe symptoms in the item “constancy of rest tremor” (F = 4.325, *p* = 0.040). These results indicate that COMT rs4680(A) is associated with more severe bradykinesia in the upper extremity and rest tremors of PD patients.

### 3.2. ALDH2 SNP rs671 (G) Is Associated with Non-Motor Symptoms but Not Associated with Motor Symptoms

We next explored the effect of ALDH2 on PD symptoms by comparing the MDS-UPDRS rating scores based on the genotypes of ALDH2 rs671. The allele frequency of rs671 (A) of the recruited patients was 23.7% (54 out of 228). The patients’ genotype frequencies were in accordance with Hardy-Weinberg equilibrium (χ^2^ = 0.041837319, *p* = 0.837930). There were no significant differences in age, gender, age of onset, levodopa equivalent daily dose (LED), entacapone use, and disease severity (Hoehn and Yahr stage, H&Y stage) between the two groups (Table 3).

With the Mann–Whitney U test, we found a significant difference in the rating scores in items “daytime sleepiness,” “lightheadedness on standing,” and “difficulty maintaining personal hygiene” (Table 4). Patients carrying rs671 (A) were found to have more severe symptoms in “daytime sleepiness,” “lightheadedness on standing,” and “difficulty maintaining personal hygiene.” With Quade’s test controlling age, sex, and LED, the group carrying rs671(A) showed a more severe problem in maintaining personal hygiene, while PD patients with genotype rs671(GG) had more severe symptoms in “depressed mood.” These results indicate that the PD patients with genotype rs671(GG) had more severe symptoms of depressed mood, while the patients carrying rs671(A) had more difficulty maintaining personal hygiene. The results also suggest that PD patients carrying rs671(A) may have more severe symptoms in “daytime sleepiness” and “lightheadedness on standing.”

### 3.3. COMT SNP rs4680 Interacts with ALDH2 SNP rs671 in Motor Symptoms of PD

To investigate COMT and ALDH2 SNPs’ interactions in the symptoms that showed statistical significance, a regression-based analysis was carried out with PROCESS macro on SPSS. Genotypes with higher enzymatic activities, rs671(GG) for ALDH2 and rs4680(GG) for COMT, were coded as 1, and genotypes with lower enzymatic activities, rs671(AG) and rs671(AA) for ALDH2 and rs4680(AG) and rs4680(AA) for COMT, were coded as 2. The genotype coding variables were mean-centered during the analysis with age, sex, and LED set as covariates to minimize confounding factors. The results are shown in Table 5.

With the linear-regression based analysis, we found that “hand movement,” “pronation-supination movement,” and “constancy of rest tremor” were significantly more severe in the PD patients carrying rs4680(A), and “depressed mood” was substantially more severe in the PD patients carrying genotype rs671(GG). Besides, the patients carrying rs671(A) reported more severe symptoms in the items of “daytime sleepiness” and “hygiene,” although there was only borderline statistical significance. More importantly, we found a significant interaction between COMT rs4680 and ALDH2 rs671 SNP in the symptom of “hand movements” (b = −1.2755, *p* = 0.02). In the PD patients with the genotype rs671 (GG), the estimated score for the “hand movement” of PD patients with rs4680 genotype (GG) was 1.8514, while the score for those carrying rs4680(A) increased significantly to 2.9431 (*p* = 0.0025). On the other hand, in the PD patients carrying rs671(A), no detrimental effect of rs4680(A) on “hand movement” was found (*p* = 0.6564), with the estimated score for those with rs4680 genotype (GG) and rs4680(A) at 2.5021 and 2.3184, respectively. PD patients carrying rs4680(A) have more severe symptoms in the hand movements, and the genotype of rs671 moderates the effect of rs4680(A). The detrimental impact of rs4680(A) is more pronounced in patients with genotype rs671(GG), while the effect of rs4680(A) is not significant in patients carrying rs671(A). A visual representation of the moderation of the effect of the COMT genotypes on hand movements by the ALDH2 genotypes is shown in Figure 1.

## 4. Discussion

This study investigated the impact of SNPs COMT rs4680(A) and ALDH2 rs671(A) on PD patients’ clinical manifestations and examined the interactions of the SNPs. The results indicate that PD patients carrying COMT rs4680(A) have a higher chance of having more severe bradykinesia in the upper extremity and rest tremor than PD patients with the rs4680(GG) genotype. The finding is compatible with our hypothesis that the reduced activity conferred by SNP rs4680(A) causes more monoamine neurotransmitters to be metabolized by MAO, resulting in increased production of DOPAL and DOPAC, both of which are neurotoxic [10,11,28,29]. Simultaneously, the reduced COMT activity also results in impaired metabolism of DOPAC [12,13]. Our finding is also supported by one study evaluating the effect of rs4680(A) on the motor symptoms and [123I]-FP-CIT binding potential of PD patients. The study showed that patients carrying rs4680(A) had significantly lower [123I]-FP-CIT binding potential, and the binding potential was shown to be correlated with motor scores [17]. The impact of rs4680(A) on PD is also evidenced by studies showing the increased risk of PD in individuals carrying rs4680(A), although ethnicity may interfere with such association [30]. At the same time, there are studies disproving the effect of rs4680(A) on PD development in the general population. However, this study indicates that rs4680(A) may cause more severe motor symptoms in PD patients, including bradykinesia in the upper extremity and resting tremor.

Besides, the present study investigated the effect of ALDH2 SNP rs671(A) or (G) on PD patients’ symptoms. We found that PD patients with genotype rs671(GG) reported a higher score for the item “depressed mood,” while the patients carrying rs671(A) reported a higher score in the item “difficulty maintaining personal hygiene.” The finding indicates that PD patients with genotype rs671(GG) are more likely to have depressed moods. Although the direct association between depression and ALDH2 rs671(G) has not been shown, the potential association of ALDH2 rs671(G) with the development of depression under pathological states may be evidenced by reports showing the interactions of ALDH2 rs671(G) with other genetic polymorphisms for the development of depression. One study found the genotype combination of ALDH2 rs671 (GG) and ADH1B rs1229984(GG) to be associated with the risk of depressive and anxiety disorders [31], and another study found a potential protective role of ALDH rs671(A) against anxiety-depressive alcohol dependence, especially in participants carrying MAOA-uVNTR 4-repeat alleles [32]. All these results suggest the potential impact of rs671(G) on developing symptoms of depressed mood. Just as these results were all observed in individuals with abnormal neuropsychological conditions, PD also involves a neurodegenerative pathology, and depression is one of the significant non-motor symptoms. Therefore, there is potential for rs671 SNP to serve as an important predictor for developing depression in PD patients, especially in East Asians.

Furthermore, PD patients carrying ALDH2 rs671(A) were found to be associated with higher scores in the item “hygiene” in this study. The results indicate the detrimental impact of rs671(A) on the motor symptoms to cause difficulty in activities to maintain personal hygiene such as washing, bathing, or brushing teeth. The result is compatible with our previous studies showing the detrimental impact of rs671(A) on the symptoms of PD [7,8]. Simultaneously, the perturbation of sleep-wakefulness regulation in PD patients carrying rs671(A) is also suggested in this study as reflected by the higher score in the item “daytime sleepiness” in the group. Although only borderline statistical significance was observed, the lack of strong association may be due to the fact that the item is not specifically designed to study the sleep-wakefulness dysregulation of PD patients. The lack of significant difference in the symptom of cognitive impairment between the two groups with different rs671 genotypes in this study may also be due to the fact that the item is not as sensitive as the Mini-Mental State Examination and Montreal Cognitive Assessment for the detection of cognitive dysfunction of PD patients [7,33]. Meanwhile, we also found that patients carrying rs671(A) reported a higher score in the item “lightheadedness on standing,” but the statistical significance was not observed after adjusting for age, sex, and LED. Future studies are needed to gain a more precise understanding of the effect of rs671(A) on the symptom.

More importantly, in the present study, in addition to the detrimental effect of COMT rs4680(A) on the motor symptoms of PD patients, we also found that ALDH2 rs671 SNP moderates the impact of COMT rs4680(A) for the symptom of “hand movements.” The negative effect of rs4680(A) on PD patients’ hand movements is more pronounced in the presence of genotype rs671(GG). This finding indicates that the reduced activity conferred by the SNP rs4680(A) secondarily causes more monoamine neurotransmitters to be metabolized by MAO, resulting in increased production of DOPAL. The simultaneous presence of fully functional ALDH2 enzyme translated by genotype rs671(GG) in the same patient causes the DOPAL to be metabolized efficiently to DOPAC. However, the reduced enzymatic activity of COMT causes inefficient metabolism and accumulation of DOPAC, and neurotoxicity ensued. Although there have not been in vitro results to prove the hypothesis, this finding suggests a novel mechanism for the interaction between COMT and ALDH2 in the pathogenesis of PD—development of neurotoxicity resulting from an imbalance in the pathway metabolizing monoamine neurotransmitter neurotoxic metabolites.

The study only recruited 114 PD patients and may be underpowered to give a comprehensive picture of the effect of the dysfunctional monoamine neurotransmitter metabolism on PD. Future studies recruiting a larger number of PD patients are warranted to validate our findings. Besides, because of the potential disruption of natural trajectories of disease progression by long-term dopamine replacement therapy, only patients with disease duration less than ten years were included. Therefore, the SNPs’ long-term impact on the clinical manifestation of PD patients is not explored in this study.

## 5. Conclusions

In conclusion, the present study examined the effect of SNPs COMT rs4680(A) and ALDH2 rs671(A) on PD symptoms. Overall, both COMT and ALDH2 inactivity and imbalance between COMT and ALDH2 lead to more severe symptoms among PD patients. The finding shed light on the importance of monoamine neurotransmitter metabolism in the pathogenesis of PD.

## Figures and Tables

**Figure 1 brainsci-11-00361-f001:**
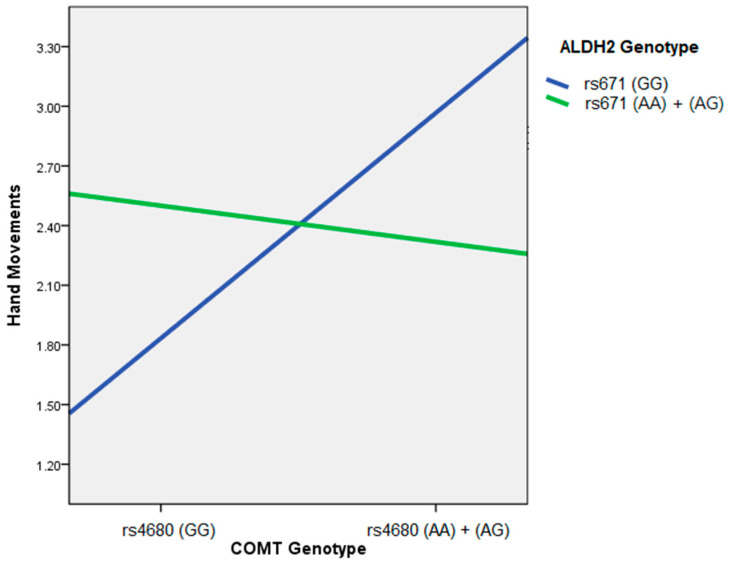
Moderation of the effect of the COMT genotypes on hand movements by the ALDH2 genotypes. In the PD patients with the genotype rs671 (GG), the estimated score for the “hand movement” of PD patients with rs4680 genotype (GG) was 1.8514, while the score for those carrying rs4680(A) increased significantly to 2.9431. On the other hand, in the PD patients carrying rs671(A), no detrimental effect of rs4680(A) on “hand movement” was found, with the estimated score for those with rs4680 genotype (GG) and rs4680(A) at 2.5021 and 2.3184, respectively.

**Table 1 brainsci-11-00361-t001:** Demographic characteristics of the study groups based on COMT rs4680 genotypes. Abbreviations: SD, standard deviation.

	GG (*n* = 62)		AG + AA (*n* = 52)		Statistic	*p*-Value
	Mean	SD	Mean	SD		
Age (years)	66.68	7.674	67.73	7.464	t = −0.739	0.462
Gender (female/male)	25/37	—	21/31	—	χ^2^ = 0.000	0.995
Age of onset (years)	61.21	8.215	62.37	7.598	t = −0.777	0.439
Levodopa equivalent dose	625.37	351.23	623.25	350.54	t = 0.032	0.975
Entacapone use (yes/no)	10/52	—	10/42	—	χ^2^ = 0.188	0.665
Stage	2.18	0.736	2.03	0.843	U = 1508.5	0.497

**Table 2 brainsci-11-00361-t002:** MDS-UPDRS results between different COMT rs4680 genotype groups.

	GG (*n* = 62)		AG + AA (*n* = 52)		Mann-Whitney U Test	*p*-Value	Quade’s Test	
	Mean	SD	Mean	SD			F	*p*-Value
Cognitive impairment	0.810	0.698	0.800	0.693	1595.000	0.913	0.032	0.859
Hallucinations and psychosis	0.190	0.507	0.370	0.747	1463.500	0.211	1.468	0.228
Depressed mood	0.630	0.891	0.590	0.829	1606.000	0.969	0.000	0.985
Anxious mood	0.530	0.646	0.630	0.720	1498.000	0.468	0.581	0.448
Apathy	0.480	0.741	0.370	0.747	1448.000	0.256	1.482	0.226
Features of DDS	0.110	0.447	0.100	0.458	1600.000	0.870	0.006	0.936
Sleep problems	0.660	1.023	0.730	1.002	1527.000	0.583	0.380	0.539
Daytime sleepiness	0.810	0.955	0.800	0.980	1592.500	0.901	0.064	0.801
Pain and other sensations	0.920	1.045	0.650	0.934	1341.500	0.092	3.277	0.073
Urinary problems	0.610	0.964	0.940	1.190	1350.500	0.097	2.843	0.095
Constipation problems	0.950	1.062	0.750	0.821	1501.000	0.498	0.552	0.459
Light headedness on standing	0.450	0.739	0.590	0.829	1438.500	0.251	1.261	0.264
Fatigue	0.790	0.852	0.750	0.868	1562.500	0.761	0.031	0.861
Speech	0.680	0.621	0.860	0.980	1473.500	0.391	1.036	0.311
Saliva and drooling	0.950	1.179	0.840	1.217	1518.500	0.557	0.457	0.500
Chewing and swallowing	0.440	0.738	0.470	0.833	1600.500	0.936	0.034	0.855
Eating tasks	0.500	0.695	0.800	0.980	1378.500	0.137	2.331	0.130
Dressing	0.950	0.858	1.040	1.076	1598.000	0.932	0.024	0.878
Hygiene	0.680	0.610	0.850	0.963	1558.500	0.737	0.293	0.590
Hand writing	0.760	0.619	0.900	0.944	1526.500	0.585	1.069	0.303
Doing hobbies and other activities	1.050	0.931	1.020	1.241	1441.500	0.305	0.905	0.344
Turning in bed	0.730	0.793	0.650	0.868	1470.000	0.374	1.026	0.313
Tremor	1.060	0.903	0.820	0.767	1355.000	0.111	2.718	0.102
Getting out of bed	1.020	0.983	1.080	1.036	1580.500	0.849	0.004	0.949
Walking and balance	1.020	1.016	0.980	1.140	1489.000	0.448	0.365	0.547
Freezing	0.500	0.805	0.450	0.879	1479.500	0.353	0.945	0.333
Speech	0.690	0.715	0.820	0.910	1553.500	0.718	0.047	0.675
Facial expression	1.310	0.822	1.410	0.779	1534.000	0.635	0.176	0.692
Rigidity	5.774	3.752	5.902	3.727	1570.500	0.813	0.004	0.947
Finger tapping	2.936	1.659	3.157	1.554	1416.500	0.257	0.712	0.401
Hand movements	2.113	1.344	2.647	1.520	1191.500	0.014 *	6.985	0.009 *
Pronation-supination movements	2.581	1.605	3.314	1.594	1220.500	0.023 *	4.225	0.042 *
Toe tapping	2.548	1.636	2.490	1.816	1586.500	0.883	0.069	0.793
Leg agility	1.677	1.513	1.980	1.594	1406.500	0.232	1.311	0.255
Arising from chair	0.350	0.680	0.510	0.880	1482.500	0.368	0.547	0.461
Gait	1.000	0.768	1.040	0.848	1598.000	0.931	0.000	0.999
Freezing of gait	0.370	0.730	0.410	0.779	1585.000	0.845	0.022	0.882
Postural stability	0.680	1.021	0.670	1.052	1568.500	0.776	0.293	0.590
Posture	1.180	0.984	1.220	1.045	1566.500	0.786	0.102	0.750
Global spontaneity of movement	0.980	0.799	1.160	0.758	1364.500	0.120	1.906	0.170
Postural tremor	0.613	0.894	0.549	0.783	1584.000	0.854	0.051	0.822
Kinetic tremor	1.290	1.165	1.431	1.171	1491.000	0.475	0.377	0.540
Rest tremor amplitude	0.645	1.202	0.863	1.200	1414.500	0.195	1.685	0.197
Constancy of rest tremor	0.610	1.136	1.060	1.448	1485.000	0.515	4.325	0.040 *

* statistical significance.

**Table 3 brainsci-11-00361-t003:** Demographic characteristics of the study groups based on ALDH2 rs671 genotypes.

	GG (*n* = 62)		AG + AA (*n* = 52)		Statistic	*p*-Value
	Mean	SD	Mean	SD		
Age (years)	67.19	7.908	67.13	7.145	t = −0.044	0.965
Gender (female/male)	24/42	—	22/26	—	χ^2^ = 1.035	0.309
Age of onset (years)	62.44	8.239	60.78	7.451	t = 1.107	0.270
Levodopa equivalent dose	589.37	382.08	672.55	295.73	t = −1.258	0.211
Entacapone use (yes/no)	12/54	—	8/40	—	χ^2^ = 0.044	0.834
Stage	2.17	0.714	2.29	0.874	U = 1431	0.311

**Table 4 brainsci-11-00361-t004:** MDS-UPDRS results between different ALDH2 rs671 genotype groups.

	GG (*n* = 66)		AG + AA (*n* = 48)		Mann-Whitney U Test	*p*-Value	Quade’s Test	
	Mean	SD	Mean	SD			F	*p*-Value
Cognitive impairment	0.750	0.662	0.880	0.733	1459.000	0.418	0.254	0.615
Hallucinations and psychosis	0.310	0.660	0.230	0.592	1487.500	0.412	0.984	0.323
Depressed mood	0.710	0.879	0.480	0.825	1319.500	0.087	5.664	0.019 *
Anxious mood	0.570	0.661	0.580	0.71	1573.000	0.944	0.458	0.500
Apathy	0.450	0.662	0.420	0.846	1461.500	0.392	2.208	0.140
Features of DDS	0.080	0.367	0.150	0.545	1524.000	0.408	0.035	0.853
Sleep problems	0.690	0.934	0.690	1.114	1486.000	0.524	1.168	0.282
Daytime sleepiness	0.660	0.940	1.000	0.968	1270.000	0.044 *	3.457	0.066
Pain and other sensations	0.710	0.980	0.920	1.028	1390.000	0.223	1.007	0.318
Urinary problems	0.750	1.104	0.770	1.057	1544.500	0.801	0	0.997
Constipation problems	0.970	1.075	0.710	0.771	1399.500	0.256	2.402	0.124
Light headedness on standing	0.380	0.678	0.690	0.879	1268.000	0.035 *	2.749	0.100
Fatigue	0.820	0.864	0.710	0.849	1463.000	0.453	1.127	0.291
Speech	0.660	0.889	0.880	0.866	1323.500	0.104	0.853	0.358
Saliva and drooling	0.940	1.261	0.850	1.072	1580.000	0.980	0.003	0.954
Chewing and swallowing	0.450	0.791	0.460	0.771	1544.000	0.779	0.068	0.794
Eating tasks	0.630	0.858	0.650	0.838	1546.500	0.801	0.125	0.725
Dressing	0.940	1.014	1.060	0.885	1391.500	0.236	0.397	0.530
Hygiene	0.620	0.804	0.960	0.771	1139.000	0.005 *	5.210	0.024 *
Hand writing	0.770	0.786	0.900	0.778	1447.500	0.379	0.174	0.677
Doing hobbies and other activities	0.970	1.015	1.130	1.160	1474.500	0.507	0.014	0.905
Turning in bed	0.650	0.818	0.750	0.838	1459.000	0.430	0.027	0.869
Tremor	0.920	0.797	1.000	0.923	1522.500	0.701	0.126	0.723
Getting out of bed	0.940	0.966	1.190	1.045	1350.500	0.153	1.101	0.296
Walking and balance	0.880	1.053	1.170	1.078	1279.000	0.058	1.909	0.170
Freezing	0.460	0.812	0.500	0.875	1560.000	0.865	0.114	0.736
Speech	0.770	0.825	0.730	0.792	1562.000	0.891	0.258	0.612
Facial expression	1.340	0.853	1.380	0.733	1537.500	0.775	0.074	0.786
Rigidity	5.969	4.031	5.646	3.297	1569.000	0.931	0.015	0.904
Finger tapping	3.062	1.580	3.000	1.663	1545.500	0.822	0	0.992
Hand movements	2.308	1.435	2.417	1.471	1542.500	0.808	0.082	0.775
Pronation-supination movements	2.939	1.609	2.875	1.684	1554.000	0.861	0.004	0.952
Toe tapping	2.431	1.639	2.646	1.816	1497.000	0.611	0.302	0.583
Leg agility	1.723	1.484	1.938	1.643	1495.500	0.603	0.490	0.485
Arising from chair	0.450	0.811	0.400	0.736	1550.000	0.812	0.279	0.599
Gait	0.920	0.797	1.150	0.799	1361.500	0.169	1.058	0.306
Freezing of gait	0.290	0.678	0.520	0.825	1319.000	0.053	2.583	0.111
Postural stability	0.690	1.045	0.650	1.021	1526.000	0.702	0.356	0.552
Posture	1.220	1.068	1.170	0.930	1571.500	0.940	0.173	0.678
Global spontaneity of movement	1.080	0.835	1.040	0.713	1545.000	0.805	0.222	0.638
Postural tremor	0.662	0.923	0.479	0.714	1471.000	0.453	0.177	0.675
Kinetic tremor	1.523	1.288	1.125	0.937	1369.500	0.201	0.642	0.425
Rest tremor amplitude	0.615	0.878	0.917	1.528	1556.500	0.856	0.193	0.661
Constancy of rest tremor	0.800	1.227	0.830	1.404	1485.00	0.515	0.042	0.839

* statistical significance.

**Table 5 brainsci-11-00361-t005:** Interaction between COMT rs4680 and ALDH2 rs671 on the symptoms affected by the SNPs.

	COMT rs4680 Main Effect			ALDH2 rs671 Main Effect			COMT rs4680 × ALDH2 rs671 Interaction Effect		
	b	SE	*p*	b	SE	*p*	b	SE	*p*
Depressed mood	−0.0332	0.1544	0.8301	−0.3134	0.1574	0.0490 *	0.5086	0.3133	0.1074
Daytime sleepiness	−0.0582	0.1729	0.7371	0.3230	0.1763	0.0698	−0.0429	0.3509	0.9030
Light headedness on standing	0.1303	0.1430	0.3641	0.2399	0.1458	0.1028	0.1502	0.2902	0.6057
Hygiene	0.1478	0.1355	0.2778	0.2729	0.1382	0.0509	0.4250	0.2750	0.1252
Hand movements	0.5547	0.2675	0.0405 *	0.0689	0.2728	0.8010	−1.2755	0.5429	0.0206*
Pronation-supination movements	0.7287	0.3015	0.0173 *	−0.0411	0.3074	0.8940	−0.0733	0.6118	0.9049
Constancy of rest tremor	0.5069	0.2464	0.0421 *	0.0384	0.2512	0.8789	0.3935	0.4999	0.4329

* statistical significance.

## Data Availability

All the associated data not provided within the paper are available on request from Chun-Hsiang Tan.

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
