# Peer review of "Interactions of COMT and ALDH2 Genetic Polymorphisms on Symptoms of Parkinson’s Disease"

_brainsci, 2021, doi:10.3390/brainsci11030361_

Round 1
Reviewer 1 Report
Review of a manuscript “Interactions of COMT and ALDH2 Genetic Polymorphisms on Symptoms of Parkinson's Disease” by Rwei-Ling Yu and coauthors submitted to the “Brain Sciences”
Parkinson's disease is a devastating neurodegenerative disease associated with by progressive denervation in the striatum and substantia nigra. These parts of the brains are essential for dopamine transmission. There is not efficient treatment of this disease preventing its development and not reliable biomarkers for early diagnostic of the disorder. The difficulty in finding a cure preventing neurodegeneration in Parkinson’s disease is explained by a complex pathogenesis with a wide arrays of symptoms, including motor and non-motor signs and heterogeneous mechanisms underlying this disorder.
The authors examined the effect of reduced activity of Catechol-O-methyltransferase and aldehyde dehydrogenase 2 conferred by single nucleotide polymorphisms COMT rs4680(A) and ALDH2 rs671(A) on the symptoms of patients with Parkinson's disease. This subject of investigation is significant and the topic of this manuscript is important. The results presented in the manuscript will be interesting for the readers of “Brain Science”.
The following corrections should be made:
Abstract:
Lines 19-20:
Monoamine neurotransmitters play essential roles in the normal functioning of our nervous system”. The sentence should be corrected as follows: ”Monoamine neurotransmitters play an essential role in the functioning of the nervous system”.
Introduction:
Lines 46-47: “…interactions between genetic susceptibility and environmental factors…”. After this sentence the authors should add the following reference:” Emamzadeh FN et al., Parkinson's disease: Biomarkers, Treatment, and Risk Factors. Frontiers in Neuroscience, 12, 61230, August 2018, https://doi.org/10.3389/fnins.2018.00612
Lines 58-60: ”Growing evidence has suggested that DOPAL and DOPAC are associated with neuronal death, and timely removal of the neurotoxic metabolites through enzymatic activity is crucial in reducing the neurotoxicity secondary to monoamine neurotransmitter metabolism.” The authors should give a reference here.
Lines 76-77:”However, results across various studies investigating the impact of the COMT SNP on individuals suffering from PD diverge”. This sentence is incomplete. The authors should explain what means this divergence and present references.
- Materials and Methods
Lines 94-103 should be deleted.
- Results
The authors should explain in the text of section “Results” clearly the data presented in Figure 1. Explanation in the legend to Figure 1 are not sufficient.
- Discussion
Lines 246-247:”We found that PD patients with genotype rs671(GG) reported a higher score for the item "depressed mood," while the patients carrying rs671(A) reported a higher score in the item "difficulty maintaining personal hygiene."
The authors should explain more clearly what they mean by “difficulty maintaining personal hygiene" and how this is related to Parkinson’s disease.
Reviewer 2 Report
Overall, I find the manuscript very novel. Furthermore, the findings of Yu et al. provide a platform for future PD research regarding COMT and ALDH2 SNP.
I have the following two minor comments,
- Authors could present the power analysis on selecting patients with the methods part.
- It would be interesting for the readers to under the relation between COMT, ALDH2, and PD if the authors could provide us their findings in a graphical representation.
Author Response
Please see the attachment.

This manuscript is a resubmission of an earlier submission. The following is a list of the peer review reports and author responses from that submission.